# Stability of SARS-CoV-2-Encoded Proteins and Their Antibody Levels Correlate with Interleukin 6 in COVID-19 Patients

Wangyang Li,[a] Georgios D. Kitsios,[a] ⓘWilliam Bain,[a] Chenxiao Wang,[b,c] Tiao Li,[a] Kristen V. Fanning,[a] Rushikesh Deshpande,[a,d] Xuebin Qin,[b,c] Alison Morris,[a] ⓘJanet S. Lee,[a] Chunbin Zou[a,d]

[a]Division of Pulmonary, Allergy, Critical Care Medicine, Department of Medicine, University of Pittsburgh School of Medicine, Pittsburgh, Pennsylvania, USA
[b]Division of Comparative Pathology, Tulane National Primate Research Center, Covington, Louisiana, USA
[c]Department of Microbiology and Immunology, Tulane University School of Medicine, New Orleans, Louisiana, USA
[d]Department of Environment and Occupational Health, University of Pittsburgh Graduate School of Public Health, Pittsburgh, Pennsylvania, USA

**ABSTRACT** The spread of severe acute respiratory syndrome coronavirus 2 (SARS-CoV-2), the causative agent of coronavirus disease 2019 (COVID-19), has become a severe global public health crisis. Therefore, understanding the molecular details of SARS-CoV-2 will be critical for fighting the virus's spread and preventing future pandemics. In this study, we globally profiled the stability of SARS-CoV-2-encoded proteins, studied their degradation pathways, and determined their correlation with the antibody responses in patient plasma. We identified 18 proteins with unstable half-lives and 6 relatively stable proteins with longer half-lives. The labile SARS-CoV-2 proteins were degraded mainly by the ubiquitin-proteasome pathway. We also observed a significant correlation between antibody levels and protein half-lives, which indicated that a stable antigen of SARS-CoV-2 could be more effective for eliciting antibody responses. In addition, levels of antiviral antibodies targeting NSP10 were found to be negatively correlated with systemic levels of interleukin 6 (IL-6) in patients. These findings may facilitate the development of novel therapeutic or diagnostic approaches.

**IMPORTANCE** SARS-CoV-2, the etiological cause of COVID-19, carries 29 genes in its genome. However, our knowledge of the viral proteins in biological and biochemical aspects is limited. In this study, we globally profiled the stability of the viral proteins in living lung epithelial cells. Importantly, the labile SARS-CoV-2-encoded proteins were mainly degraded through the ubiquitin-proteasome pathway. Stable proteins, including spike and nucleocapsid, of SARS-CoV-2 were more effective in eliciting antibody production. The levels of antiviral antibodies targeting NSP10 were negatively correlated with systemic levels of IL-6 in COVID-19 patients.

**KEYWORDS** SARS-CoV-2, COVID-19, protein degradation, ubiquitination, antiviral antibody, interleukin 6, IL-6

Severe acute respiratory syndrome coronavirus 2 (SARS-CoV-2), the causative agent of the coronavirus disease 2019 (COVID-19) pandemic, has created a global health crisis (1). The high transmissibility and global disease burden of COVID-19 have led to an extraordinary collective effort of researchers to understand the molecular biology of SARS-CoV-2 (2–6). SARS-CoV-2 is an enveloped, single-stranded, positive-sense RNA betacoronavirus belonging to the family *Coronaviridae* of the order *Nidovirales* (7). The genome of SARS-CoV-2 is about 30 kb in length, containing 10 open reading frames (ORFs), among which ORF1ab encodes the replicase polyproteins PP1ab and PP1a, cleaving them into 16 nonstructural proteins (NSPs), and the remaining ORF2 to -10 encode 4 viral structural proteins, spike (S), nucleocapsid (N), membrane (M), and envelope (N), and 8 accessory proteins (8). Broadly, these proteins are involved in complex biological processes, including RNA replication, epigenetic and gene expression regulation, vesicle trafficking,

**Ad Hoc Peer Reviewer** ⓘİkbal Agah İnce, University Medical Center Groningen - UMCG

Address correspondence to Chunbin Zou, zouc@upmc.edu.

The authors declare no conflict of interest.

lipid modification, RNA processing and regulation, ubiquitin ligases, nuclear transport machinery, the cytoskeleton, mitochondria, and extracellular matrix signaling (9, 10). Based on their functions and interactions with host cells, these proteins are grouped into three categories: host entry, self-acting, and host defense (10–15). Therefore, understanding the mechanism of SARS-CoV-2 interaction with host cells could guide the development of effective therapeutic strategies or diagnostic approaches.

Proteins are functional macromolecules that are dynamic in living mammalian cells (16). SARS-CoV-2 makes use of the host translational machinery to synthesize viral proteins to assemble and replicate viral particles in host cells (7, 17). It is conceivable that the virus-encoded proteins may be subject to posttranslational modification and regulation by host cells. Ubiquitination, an important and ubiquitous form of protein posttranslational modification, plays a crucial role in controlling the stability of targeted proteins by regulating their intracellular degradation (18). The stability of cellular proteins is highly variable, from a few minutes to several hours, and can be tightly regulated in responding to various external and internal pathophysiological inputs (19). Ubiquitination of viral proteins thus leads to subsequent protein degradation. Our understanding of viral protein ubiquitination and subsequent degradation is limited. In addition, the pathophysiological significance of viral protein ubiquitination and degradation is not fully understood in SARS-CoV-2.

In the present study, our primary objectives were to (i) globally profile the stability of SARS-CoV-2 proteins and their degradation pathways, (ii) determine whether the stability of SARS-CoV-2 proteins correlates with antibody responses, and (iii) study if virus-specific antibodies are correlated with proinflammatory interleukin 6 (IL-6) levels in COVID-19 patients, which may aid in the development of effective therapeutic and diagnostic approaches.

## RESULTS

**SARS-CoV-2-encoded proteins are dynamic *in vivo*.** Since viral proteins are synthesized in host cells, they may be dynamic like the cellular proteins. Previous studies have reported that the total mRNA of the SARS-CoV-2 *N* (nucleocapsid) gene reaches its highest level by 3 days postinfection (dpi) ($10^9$) from 1 dpi ($10^5$) and then decreases by 6 dpi ($10^6$) based on subgenomic nucleocapsid RNA copies per 100 ng total tissue (20). To verify the above-mentioned observation, we infected *K18-hACE2*$^{+/-}$ and nontransgenic *K18-hACE2*$^{-/-}$ mice with SARS-CoV-2 (20). Lung tissues were examined with anti-spike antibody by immunofluorescence. As expected, anti-spike immunofluorescence was minimally observed in the nontransgenic *K18-hACE2*$^{-/-}$ mouse lung tissues, suggesting that the anti-spike antibody was spike protein specific (Fig. 1A, top). The fluorescence intensity of the spike protein (green) was markedly increased at 3 dpi in *K18-hACE2*$^{+/-}$ mice (Fig. 1A, middle). However, the fluorescence intensity did not markedly decrease at 6 dpi (Fig. 1B). This observation suggested that the expression of viral proteins is dynamic and may not always correspond to their RNA levels in mouse lung tissues.

**Determining the stability of SARS-CoV-2 proteins.** To study the stability of viral proteins, we subcloned the predicted SARS-CoV-2 proteins. The SARS-CoV-2 genome encodes 29 identifiable proteins or polypeptides (Fig. 2A). All SARS-CoV-2 protein constructs were cloned into a mammalian expression vector containing a C-terminal V5 epitope tag (within the pcDNA3.1 plasmid) or 2×Strep-tag II (within the pLVX-EF1α-IRES-Puro plasmid). The accuracy of the subcloned viral genes in the plasmids was confirmed by sequencing. The plasmids containing individual viral genes were transduced into BEAS-2B cells, and expression was verified by immunoblot analysis. The expression of 25 out of 29 SARS-CoV-2 proteins was successfully verified by observing immunoblot results consistent with the predicted protein sizes (Fig. 2B). NSP11, ORF7a, ORF9c, and ORF10 were not detected by immunoblot analysis, possibly due to their smaller molecular sizes. Despite the failure to detect NSP3 by Western blotting, we selected NSP3d, a domain of the protein named papain-like protease (PL$^{pro}$).

Given the dynamic nature of viral protein expression, we measured the half-lives of the SARS-CoV-2 proteins. Immunoblot analysis of the viral proteins was conducted (see Fig. S1, Table S3 in the supplemental material). The densitometric results of the immunoblot analysis normalized to β-actin were plotted, and the half-lives of 25 proteins were

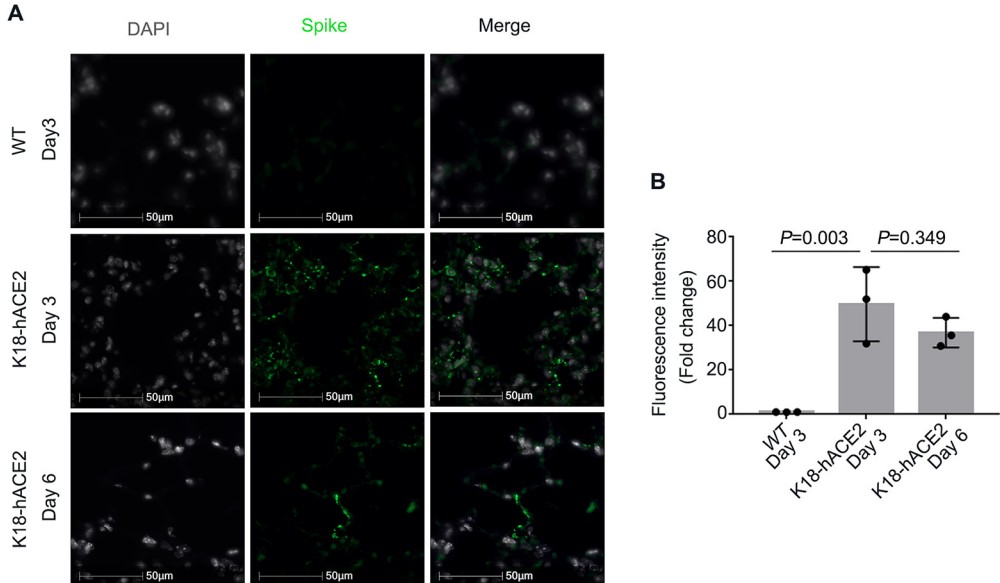

**FIG 1** SARS-CoV-2-encoded proteins are dynamic *in vivo*. (A) Immunofluorescence staining of spike protein in the lung tissues of SARS-CoV-2-infected mice. *K18-hACE2*$^{-/-}$ mice or *K18-hACE2*$^{+/-}$ transgenic mice were infected with SARS-CoV-2 for 3 and 6 days. Mouse lung sections were examined with SARS-CoV-2 spike antibody (green) and DAPI (white) by immunofluorescence staining. (B) Relative densitometry of spike. WT, wild type.

determined (Fig. 3). Among them, 18 unstable proteins (NSP1, NSP3d, NSP4, NSP6, NSP7, NSP8, NSP9, NSP12, NSP13, NSP14, NSP16, E, ORF3a, ORF3b, ORF6, ORF7b, ORF8, and ORF9b) had short half-lives within the range of 0.4 to 8 h (Fig. 3A and B). Among them, NSP1, E, and ORF8 have half-lives of 1.57 h, 3.92 h, and 0.48 h, respectively (Fig. 3A). The remaining 7 proteins (NSP2, NSP5, NSP10, NSP15, spike, N, and M) were relatively stable, with long half-lives of over 8 h (Fig. 3C). These results showed that 18 of the studied viral proteins were unstable and that 7 of them had long half-lives.

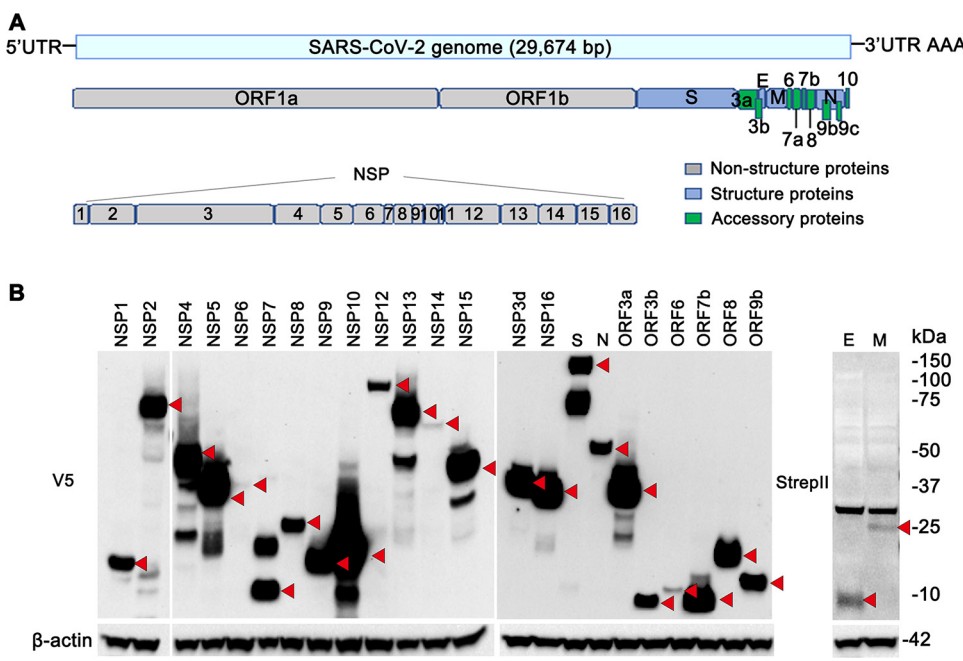

**FIG 2** Cloning of SARS-CoV-2 genes. (A) Schema of the SARS-CoV-2 genome. UTR, untranslated region. (B) The expression of 25 out of 29 SARS-CoV-2 proteins examined was verified by immunoblotting (indicated with red arrowheads). The SARS-CoV-2 genes were subcloned into the pcDNA3.1-V5-His vector and transfected into BEAS-2B cells. Cell lysates were collected and immunoblotted with V5, Strep-tag II, and $\beta$-actin antibodies.

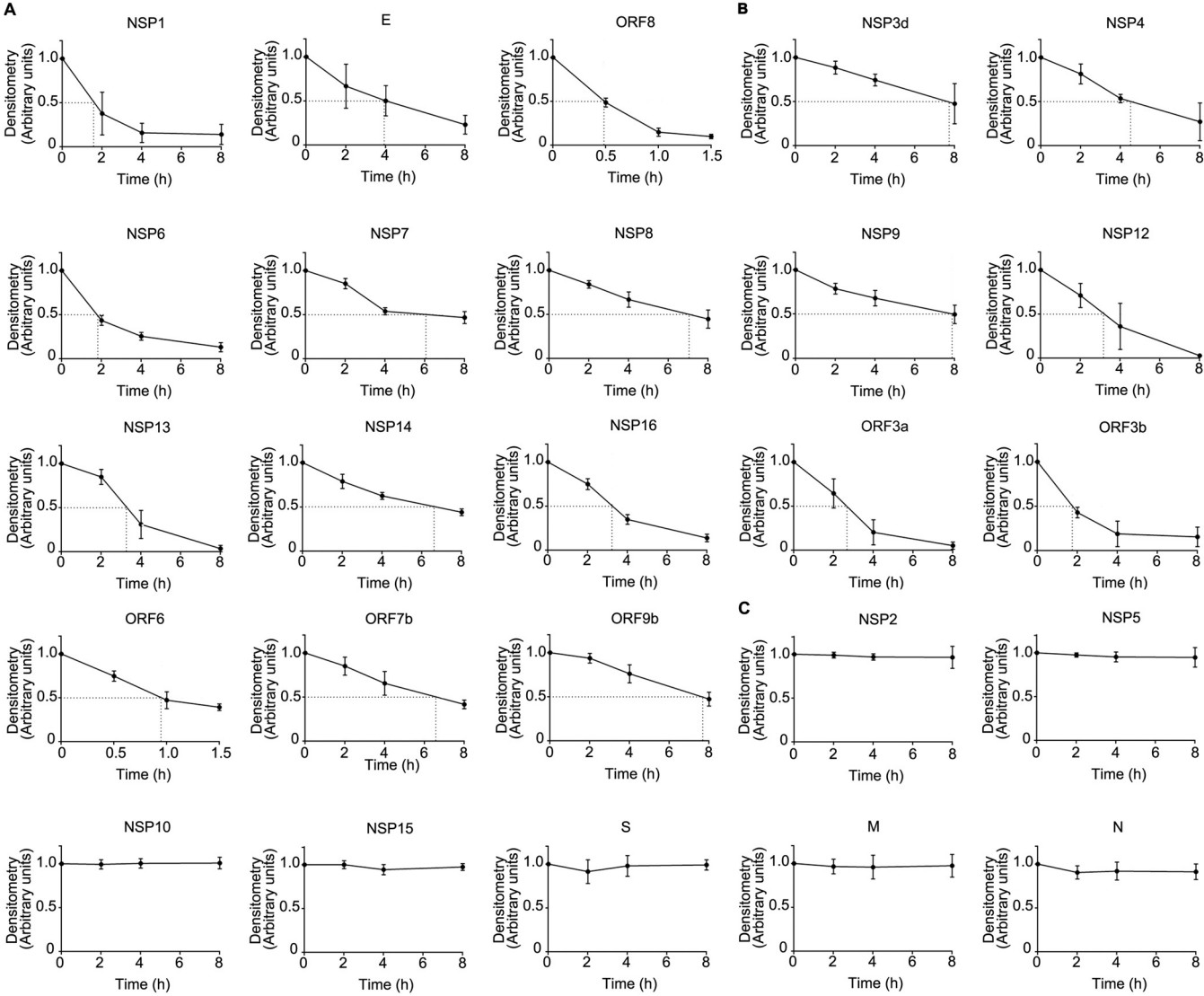

**FIG 3** Determining the stability of SARS-CoV-2 proteins. Human lung epithelial BEAS-2B cells were exposed to cycloheximide (CHX) (100 μg/mL) at four time points (2, 4, 6, and 8 h) prior to immunoblotting. The bands were quantitated and normalized to β-actin, and the half-lives of the indicated proteins are shown graphically. (A) NSP1, E, and ORF8. (B) NSP3d, NSP4, NSP6, NSP7, NSP8, NSP9, NSP12, NSP13, NSP14, NSP16, ORF3a, ORF3b, ORF6, ORF7b, and ORF9b. (C) NSP2, NSP5, NSP10, NSP15, spike, M, and N. Data are from three independent biological replicates.

**Labile SARS-CoV-2 proteins were degraded via the proteasome system.** In eukaryotic cells, the ubiquitin-proteasome system is believed to be the major player in maintaining proteostasis. To investigate if the ubiquitin-proteasome degradation pathway is involved in the degradation of viral proteins, we examined protein stability in the presence or absence of the proteasome inhibitor MG132 (20 μM) or the lysosome inhibitor leupeptin (10 μM) along with cycloheximide (CHX) (Fig. S2). Interestingly, the decreased expression of all of the unstable proteins (NSP1, NSP4, NSP6, NSP7, NSP8, NSP9, NSP12, NSP13, NSP14, NSP16, ORF3a, and ORF8) induced by CHX could be effectively reversed by MG132 treatment (Fig. 4A), except for NSP3d (Fig. 4B). However, leupeptin exposure did not affect the stability of the unstable viral proteins, indicating that the degradation of these proteins is mediated mainly through the proteasomal pathway rather than the lysosomal pathway (Fig. 4B). As an internal control, the addition of MG132 or leupeptin to CHX-treated cells did not affect the spike protein level, which was consistent with our observation (Fig. 3C) that spike is a stable protein with a longer half-life in lung epithelial cells (Fig. 4C). These data indicated that the unstable proteins are degraded mostly via ubiquitin-proteasome pathways.

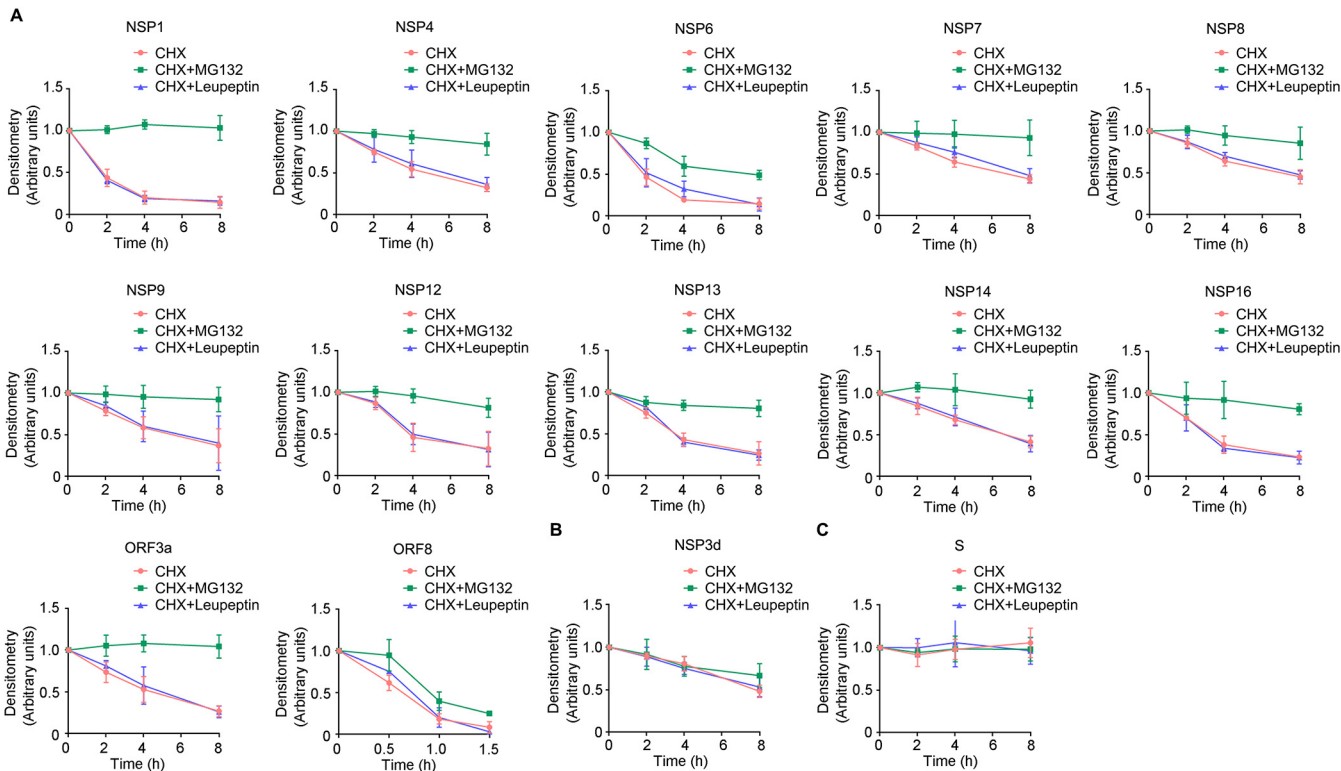

**FIG 4** Labile SARS-CoV-2 proteins are degraded via the proteasome system. Equivalent amounts of pcDNA3.1-V5 plasmids (2 μg) were transfected into BEAS-2B cells for 48 h. Cells were treated with cycloheximide (CHX), CHX with MG132, and CHX with leupeptin for various times (2, 4, 6, and 8 h). Cell lysates were subjected to V5 and β-actin immunoblotting. The bands were quantitated and normalized to β-actin. Densitometry results for each group are plotted. (A) NSP1, NSP4, NSP6, NSP7, NSP8, NSP9, NSP12, NSP13, NSP14, NSP16, ORF3a, and ORF8. (B) NSP3d. (C) Spike. Data are from three independent biological replicates.

**Ubiquitination of labile SARS-CoV-2 proteins.** Ubiquitination is an important form of protein posttranslational modification and plays a crucial role in controlling the stability of target proteins by regulating their intracellular degradation. To determine whether ubiquitin is sufficient to trigger the labile viral proteins for degradation, we cotransfected plasmids containing viral genes with hemagglutinin (HA)-tagged ubiquitin plasmids in cells, followed by immunoblot analysis. Ubiquitin was successfully expressed in cells, and overexpressed HA-ubiquitin plasmids promoted the protein instability of NSP1, NSP6, NSP7, NSP8, NSP9, NSP12, NSP13, NSP14, NSP16, ORF3a, and ORF8 (Fig. 5A). The protein instability exhibits a ubiquitin plasmid concentration-dependent manner. As expected, excessive ubiquitin did not influence the expression of the spike protein (Fig. 5B).

To confirm whether these proteins are polyubiquitinated, we performed further analysis by immunoprecipitation (IP) where the cell lysates were subjected to immunoprecipitation after transfection with V5-tagged plasmids, followed by immunoblotting of V5 immunoprecipitates with ubiquitin antibody. In these studies, ubiquitin was bound to the NSP1, NSP9, and ORF3a proteins in immunoprecipitates in a pattern consistent with the polyubiquitinated substrate (Fig. 6A to C). As a stable protein, co-IP of ubiquitin with V5-tagged spike protein failed to show specific binding above the level of the control, suggesting that the spike protein might be devoid of a ubiquitin acceptor site (Fig. 6D). These data indicate that SARS-CoV-2-encoded unstable proteins are mostly polyubiquitinated and that ubiquitin is required for their protein lability.

**SARS-CoV-2 protein-specific antibodies in COVID-19 patients.** As mentioned above, we identified seven stable SARS-CoV-2 proteins, NSP2, NSP5, NSP10, NSP15, spike, M, and N. To globally profile the antibody response against these seven proteins from the plasma of COVID-19 patients, we collected plasma samples from 19 COVID-19 patients and determined the corresponding IgG antibody titers using an indirect enzyme-linked

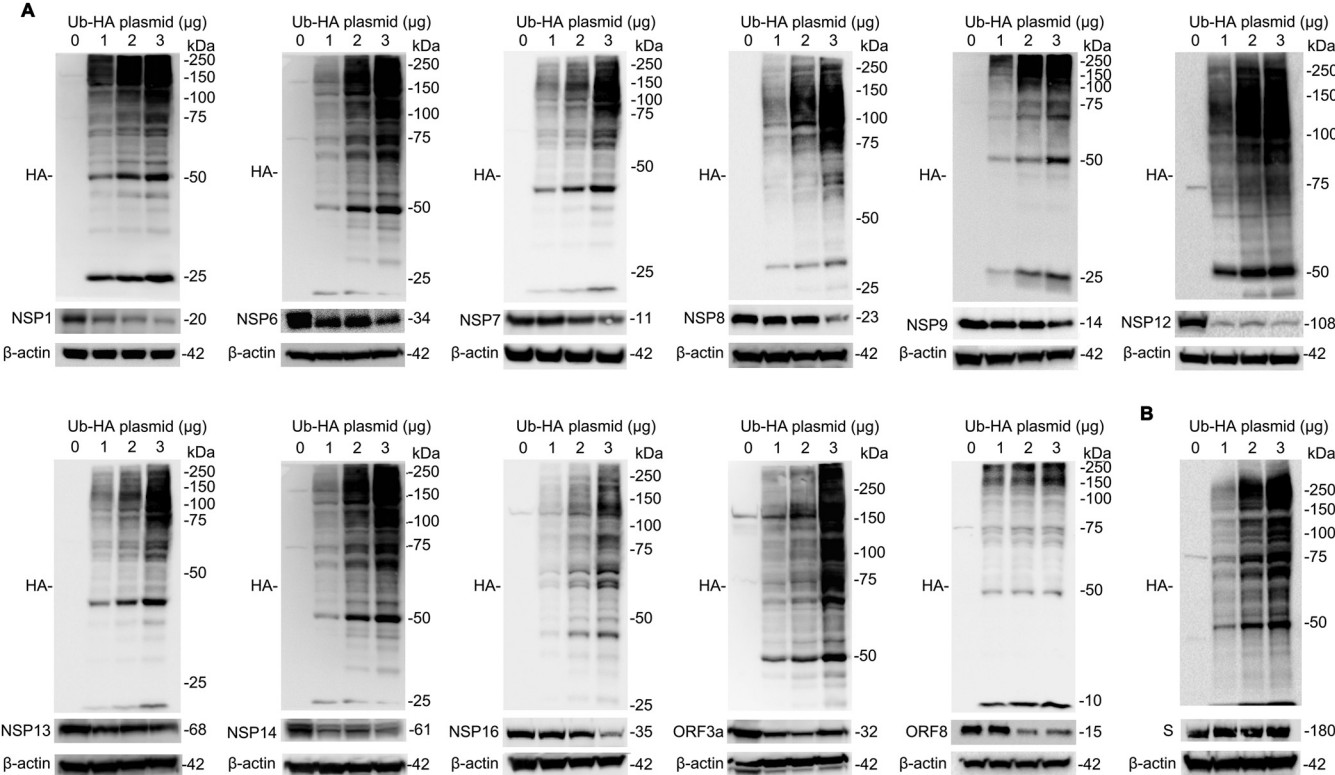

**FIG 5** Ubiquitin impairs the stability of SARS-CoV-2 proteins. Various ubiquitin (Ub)-HA plasmids (1, 2, and 3 $\mu$g) were transfected into BEAS-2B cells. Cell lysates were immunoblotted with HA antibody and the indicated antibodies against SARS-CoV-2 proteins. (A) NSP1, NSP6, NSP7, NSP8, NSP9, NSP12, NSP13, NSP14, NSP16, ORF3a, and ORF8. (B) Spike. Data are representative of results from three independent experiments.

immunosorbent assay (ELISA). Plasma samples from non-COVID-19 patients were used as controls. The recombinant SARS-CoV-2-encoded proteins used in the ELISA were commercially purchased. Among the tested proteins, levels of IgGs against spike, N, NSP2, and NSP15 were high overall (Fig. 7A). However, the specificities of the IgGs against NSP2 and NSP15 were low, without statistical significance; their specificity may be improved by antigen selection used in the assay (Fig. 7A). Levels of IgGs against E, M, NSP1, NSP5, NSP10,

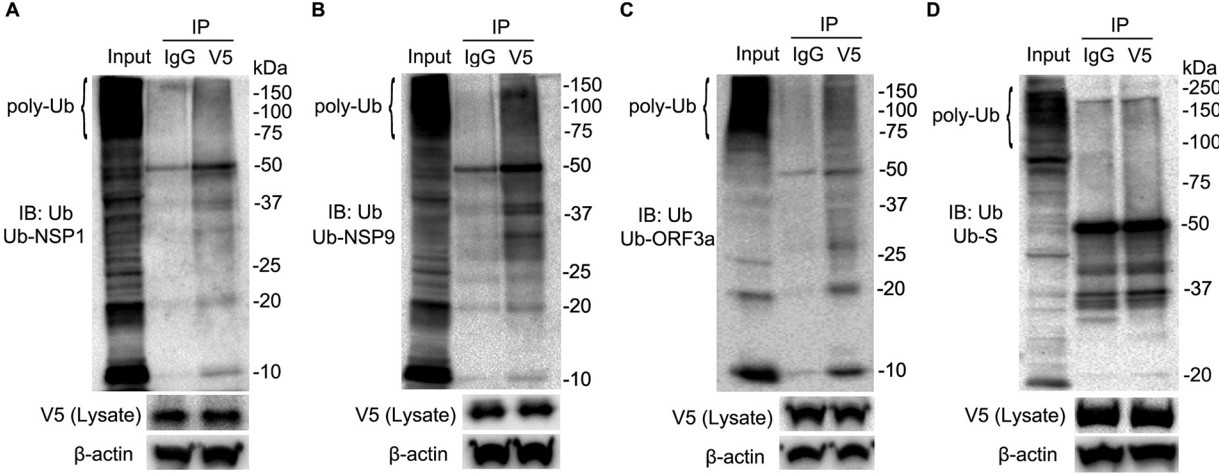

**FIG 6** Ubiquitination of labile SARS-CoV-2 proteins. The pcDNA3.1-V5 plasmids (2 $\mu$g) carrying the indicated SARS-CoV-2 genes were transfected into BEAS-2B cells for 48 h. BEAS-2B cells were pretreated with MG132 for 1 h and then lysed with IP buffer with 5 $\mu$M ubiquitin aldehyde. Thereafter, the cell lysates were subjected to immunoprecipitation, followed by ubiquitin (Ub) and V5 immunoblotting (IB). (A) NSP1. (B) NSP9. (C) ORF3a. (D) Spike. Poly-Ub denotes polyubiquitin chains. Data are representative of results from three independent experiments.

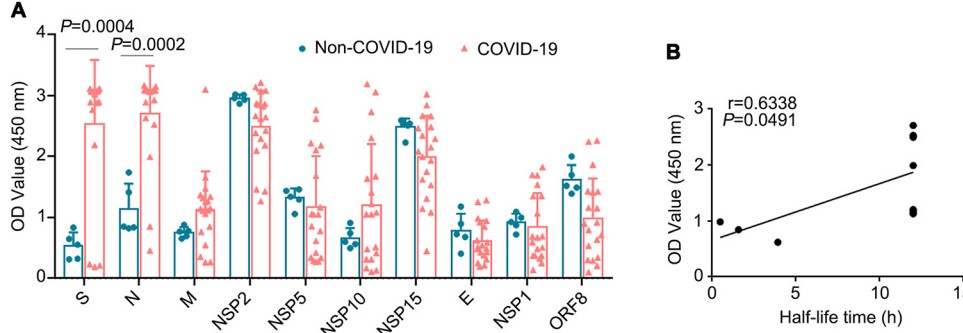

**FIG 7** SARS-CoV-2 protein-specific antibodies in COVID-19 patients. (A) Plasma samples from 19 COVID-19 patients and 5 healthy controls were collected. The recombinant SARS-CoV-2 proteins, including NSP2, NSP5, NSP10, NSP15, spike, M, E, and N, were used to determine the corresponding IgG antibody titers in COVID-19 patients using an indirect ELISA. (B) Correlation of OD values (450 nm) and half-lives of proteins in COVID-19 patients. The half-lives of stable NSP2, NSP5, NSP10, NSP15, spike, N, M, and E proteins and their corresponding antibody levels were analyzed. IgG titers are indicated with OD values (at 450 nm).

and ORF8 were relatively low; among them, E, NSP1, and ORF8 were short-life proteins. We hypothesized that the stability of the antigen may alter the production of the corresponding antibodies. We determined whether the half-lives of the SARS-CoV-2-encoded proteins (seven stable proteins [NSP2, NSP5, NSP10, NSP15, S, M, and N] and three unstable proteins [E, NSP1, and ORF8]) correlated with specific antibody responses. Linear regression analysis suggested that the stability of the proteins may be positively correlated with their IgG levels in the plasma of COVID-19 patients (Fig. 7B). These data suggested that both the stable viral spike and N proteins produced high levels of specific antibodies.

**Correlation of antiviral antibodies and the proinflammatory mediator IL-6 in COVID-19 patients.** Infection by SARS-CoV-2 induces antibody production, activates effector T lymphocytes, and induces a proinflammatory response. A key proinflammatory mediator is the cytokine IL-6, which is linked to the severity and prognosis of COVID-19 (21–23). To understand the potential influence of virus-specific antibodies on inflammation, we determined the correlation between the levels of viral protein-specific antibodies and the proinflammatory mediator IL-6 (Fig. 8). Interestingly, among the antibodies analyzed, antibody against NSP10 protein may be negatively correlated with IL-6 (Fig. 8A). Antibodies against other viral proteins, spike, E, N, M, NSP1, NSP2, NSP5, NSP15, and ORF8, may not be significantly correlated with IL-6. These data suggested that SARS-CoV-2-specific antibodies were involved in COVID-19-associated inflammation and that the NSP10 protein may be crucial for IL-6 generation and secretion.

## DISCUSSION

In this study, we (i) globally profiled the stability of 25 proteins (out of 29 total predicted proteins) encoded by SARS-CoV-2, (ii) identified that the labile SARS-CoV-2-encoded proteins were degraded mainly through the ubiquitin-proteasome pathway, (iii) determined that the stable proteins of SARS-CoV-2 corresponded to higher levels of IgG production, and (iv) determined that SARS-CoV-2-specific antibodies were correlated with the proinflammatory mediator IL-6 in COVID-19 patients.

Human lung bronchial epithelial BEAS-2B cells were used to study the half-lives of the viral proteins since SARS-CoV-2 first invades airway epithelial cells and then disseminates to other organs. SARS-CoV-2 encodes 29 predicted proteins; 25 viral proteins were successfully subcloned and expressed in BEAS-2B cells. Proteins play a central role in virtually every biological process, including proteins involved in gene regulation, transcription, translation, processing, and the secretion of other proteins (24, 25). The SARS-CoV-2 genome encodes 29 confirmed proteins, which mediate virus invasion, facilitate the viral replication cycle and self-assembly, and hijack the host proteasome and assist in host immune escape (7, 9, 26). Recently, SARS-CoV-2 NSP1 has been shown to inhibit host protein synthesis by placing its C-terminal domain into the host ribosomal mRNA channel and to facilitate efficient viral replication and immune evasion by blocking innate immune responses (27, 28). Open

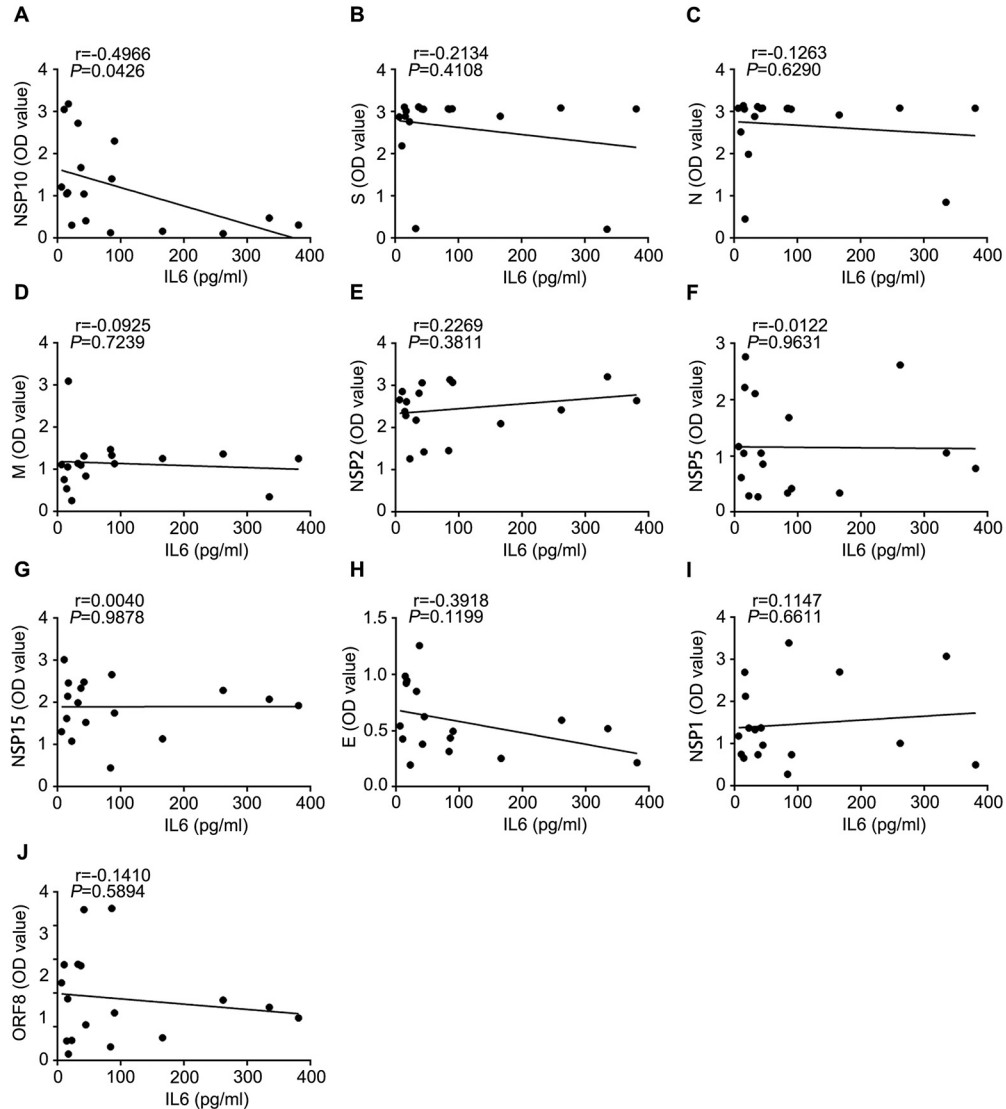

**FIG 8** Correlation of plasma IL-6 concentrations in COVID-19 patients and SARS-CoV-2 protein-specific antibodies. (A) Correlation between IL-6 and NSP10. (B to J) Correlation between IL-6 and spike, N, M, NSP2, NSP5, NSP15, E, NSP1, and ORF8, respectively.

reading frame 3a (ORF3a), the largest accessory protein of SARS-CoV-2, has been reported to protect SARS-CoV-2 from being cleared by inhibiting the fusion of autophagosomes with lysosomes in host cells (29) and to induce cell apoptosis in a membrane-dependent manner (30). SARS-CoV-2 nonstructural proteins assemble a replication-transcription complex (RTC) (31). NSP12 RNA-dependent RNA polymerase, coupled with the helicase NSP13, forms a SARS-CoV-2 mini-replication-transcription complex in the virus life cycle (31, 32). Understanding which individual SARS-CoV-2 proteins are degraded can provide important clues on how we might interfere with these processes and, consequently, which of those proteins can be targeted for antiviral therapeutics (9–11).

Protein abundance reflects the balance of synthesis and degradation (33). Since the viral proteins were dynamic, we studied the stability of the viral proteins. According to their half-lives, these viral proteins can be classified into two groups: 18 unstable proteins and 7 relatively stable proteins with longer half-lives of more than 8 h. Interestingly, most of the unstable proteins were degraded via the ubiquitin-proteasome-mediated degradation pathway that mainly degrades short-lived regulatory proteins and soluble misfolded proteins and polypeptides (34). ORF8 is one of the short-life proteins, with a half-life of about

30 min. ORF8 binds to major histocompatibility complex (MHC) class I molecules for lysosomal degradation, thus evading immune surveillance (35). The proteasome degrades ubiquitinated proteins, and ubiquitination is a cascade of enzymatic reactions, although the E3 ubiquitin ligases that govern viral protein degradation remain to be discovered (36, 37).

Protein stability has been demonstrated to be a predominant parameter in the control of antigen processing, thereby determining immunogenicity, immune responses, and, eventually, antibody production (38, 39). Prolonged antigen presentation maintains interactions among T cells and dendritic cells and consequently enhances follicular helper T (Tfh) cell differentiation, which boosts antibody production by B cells to form long-lasting immunity against virus infection (40). We show that SARS-CoV-2 proteins exhibit distinct half-lives and elicit various degrees of antibody responses. The spike and N proteins have been widely used as antigens for the diagnosis of COVID-19 due to their high immunogenicity (41, 42). Consistent with this observation, our data showed that both spike protein and N protein possess the highest levels of antibody responses among the viral proteins. Preexisting memory T cell immunity to common cold coronaviruses in humans can prime the response to SARS-CoV-2 (43, 44). Therefore, antibody cross-reactivity might result in high levels of NSP2 and NSP15 antibodies in healthy controls, as shown in our study. Developing conserved epitopes of NSP2- and NSP15-specific antibodies for SARS-CoV-2 infection could improve positive diagnostic rates. Overall, our study provides a systemic view of these vital antibody responses, in addition to profiling SARS-CoV-2 protein stability.

Hyperinflammation is a prominent feature in COVID-19 patients (45). The cytokine IL-6 is a key proinflammatory mediator that is linked with disease severity and poor prognosis in COVID-19 patients (21, 22). Multiple types of cells secrete IL-6 after viral products bind to the surface receptors of the cells (21, 46). Whether a specific SARS-CoV-2-encoded protein(s) can stimulate the production of IL-6 is not fully understood. Theoretically, viral products activate B lymphocytes to produce specific antibodies or activate effector T cells, and the synergistic action of both antibodies and effector T cells terminates the toxicity of the viral products (47). The confluence of viral products may alter the corresponding antibody levels. In response, antibodies may neutralize the corresponding viral products and thereby decrease the availability of the viral products that induce cells to secrete proinflammatory mediators (48). In this study, we assayed the correlation between viral protein-specific antibodies and the inflammatory mediator IL-6. We identified that anti-NSP10 antibody may be negatively correlated with plasma IL-6 levels, in contrast to other SARS-CoV-2 proteins. The intrinsic link between specific antiviral antibodies and systemic inflammation is complex and warrants further investigation in the future.

Taken together, we profile a comprehensive SARS-CoV-2 protein half-life landscape, in which the unstable proteins are polyubiquitinated and degraded mainly through the ubiquitin-proteasome system and stable proteins of SARS-CoV-2 are more effective in eliciting IgG responses. In addition, we identified that the plasma level of anti-NSP10 antibody is negatively correlated with systemic IL-6 levels. The findings in this study may facilitate the development of potential therapeutic or diagnostic approaches against SARS-CoV-2 infection.

## MATERIALS AND METHODS

**Cell line and reagents.** Human lung epithelial BEAS-2B cells were cultured in HITES (500 mL DMEM/F12, 2.5 mg insulin, 2.5 mg transferrin, 2.5 mg sodium selenite, 2.5 mg transferrin, 10 $\mu$M hydrocortisone, 10 $\mu$M Beta-estradiol, 10 mM HEPES, 2 mM L-glutamine) medium containing 10% fetal bovine serum (FBS) and maintained at 37°C in a humidified atmosphere of 5% $CO_2$ as previously described (49). Cycloheximide, MG132, and leupeptin were obtained from Calbiochem (Billerica, MA). Hemagglutinin (HA) tag (1:1,000) (catalog number 3724), $\beta$-actin (1:1,000) (catalog number A5441), and ubiquitin (1:1,000) (catalog number 3933S) antibodies were purchased from Cell Signaling Technology (Danvers, MA). SARS-CoV-2 ORF8 antibody (1:1,000) (catalog number GTX135591) was purchased from GeneTex (Irvine, CA). SARS-CoV-2 spike S1 antibody (1:5,000) (catalog number 40150-R007) and spike antibody (1:1,000) (catalog number 3425) were obtained from Sino Biological (San Diego, CA) and ProSci (Poway, CA), respectively. V5 tag antibody (1:5,000) (catalog number 46-1157), mouse anti-human IgG(H+L) secondary antibody (1:5,000) (catalog number 31420), the pcDNA3.1-V5-His-TOPO cloning kit (catalog number K490001), and TOP10 competent cells (catalog number C404006) were obtained from Invitrogen (St. Louis, MO). Anti-Strep-tag II antibody (1:1,000)

(catalog number ab76949) and 3,3′,5,5′-tetramethylbenzidine (TMB) ELISA substrate (highest sensitivity) (catalog number ab171522) were purchased from Abcam (Cambridge, MA). A goat anti-mouse IgG-horseradish peroxidase (HRP) conjugate (1:5,000) (catalog number 1706516) and a goat anti-rabbit IgG-HRP conjugate (1:5,000) (catalog number 1706516) were obtained from Bio-Rad (Hercules, CA). Goat anti-mouse IgG(H+L) (Alexa Fluor 488) (1:500) (catalog number ab150113) was obtained from Abcam (Waltham, MA). Ubiquitin aldehyde (1:150) (catalog number BML-UW8450-0050) was obtained from Enzo Life Sciences (Farmingdale, NY). A complete protease inhibitor cocktail (catalog number 88266) was purchased from Pierce (Rockford, IL). ZR plasmid miniprep classic was obtained from Zymo Research (Irvine, CA). Recombinant SARS-CoV-2 NSP1 protein (catalog number 81314), recombinant SARS-CoV-2 NSP2 protein (catalog number 81323), and recombinant SARS-CoV-2 NSP5 protein (catalog number 81320) were obtained from Active Motif (Carlsbad, CA). Recombinant SARS-CoV-2 nucleocapsid protein (catalog number 40588-V08B) and recombinant SARS-CoV-2 spike S1 protein (catalog number 40591-V08H) were purchased from Sino Biological (San Diego, CA). Recombinant SARS-CoV-2 NSP10 protein (catalog number 10-408) and recombinant SARS-CoV-2 ORF8 protein (catalog number 10-436) were purchased from ProSci (Poway, CA). Recombinant SARS-CoV-2 NSP15 protein (catalog number 89912), recombinant SARS-CoV-2 membrane protein (catalog number 230-01124), and recombinant SARS-CoV-2 envelope protein (catalog number RP-87682) were purchased from MyBioSource (San Diego, CA), RayBiotech (Peachtree Corners, GA), and Thermo Fisher Scientific (Waltham, MA), respectively. Coating buffer (pH 9.6) (catalog number c3041-50CAP) and phosphate-buffered saline (PBS) (pH 7.4) (catalog number P4417-100TAB) were purchased from Sigma (Bronx, NY). Tris-EDTA buffer (pH 9.0) (catalog number ab93684) was obtained from Abcam. The SARS-CoV-2 plasmids were obtained from Addgene, and detailed information is shown in Table S1 in the supplemental material.

**Molecular cloning of the viral genes.** The directional TOPO cloning procedure was conducted according to the manufacturer's instructions. The full-length cDNAs of SARS-CoV-2 plasmids were used as the templates to amplify the genes of interest. Primer sequences for the genes are listed in Table S2. The reaction was performed at 95°C for 30 s, 45°C to 72°C (according to the melting temperature [$T_m$]) for 30 s, and 72°C for 1 min/kb with 25 cycles in a PCR detection system (Bio-Rad, Hercules, CA). The expected length of blunt-end PCR products was subsequently checked on a 1% agarose gel, and the products were ligated into pcDNA3.1D/V5-His-TOPO for 5 min at room temperature. Next, 5 $\mu$L of the reaction mixture was transformed into TOP10 chemically competent *Escherichia coli* cells. One hundred microliters from each transformation mixture was spread onto an antibiotic-containing LB agar plate. Single colonies were picked up and inoculated for plasmid preparation. The subcloned genes were verified by sequencing with the T7 primer, and the expression of cDNA-encoded proteins was determined by Western blot analysis using antibody for the V5 epitope of the fusion proteins.

**Plasmid transfection.** Human bronchial epithelial BEAS-2B cells were transfected with the indicated plasmids by electroporation. Briefly, 1 million cells were suspended in 100 $\mu$L of transfection buffer (20 mM HEPES in PBS buffer) in a 2-mm electroporation cuvette and transfected using nucleofection (nucleofected) with 2 $\mu$g of expression plasmids using a nuclear transfection apparatus (Amaxa Biosystem, Gaithersburg, MD) with preset program T-013 in the Nucleofection II system. The cells were then transferred into six-well plates and cultured in 2 mL HITES medium for 48 h.

**Western blotting.** Immunoblotting was performed as previously described (49). Briefly, cells were collected and lysed with cold lysis buffer (1× PBS, 0.5% Triton X-100, and a protease inhibitor cocktail). The cell lysates were cleared by centrifugation at 13,000 rpm for 10 min at 4°C after being subjected to sonication, followed by the determination of protein concentrations using a bicinchoninic acid (BCA) assay kit. Proteins were separated by sodium dodecyl sulfate-polyacrylamide gel electrophoresis (SDS-PAGE) and then transferred onto a nitrocellulose membrane. After being blocked with 5% (wt/vol) nonfat dry milk in Tris-buffered saline containing 0.1% Tween 20 (0.1% TBS-T), membranes were incubated with the appropriate dilutions of specific primary antibodies overnight at 4°C. HRP-conjugated secondary antibodies were applied at a 1:5,000 to 1:10,000 dilution for 1 h at room temperature. The bands were visualized with enhanced chemiluminescence (ECL) and quantified using a Bio-Rad Ultrachemi imaging system.

**Coimmunoprecipitation.** For coimmunoprecipitation (co-IP), cells were lysed with IP lysis buffer (1× PBS, 0.5% Tween 20, and a protease inhibitor cocktail) at 4°C. The lysates were cleared by centrifugation at 13,000 rpm for 10 min at 4°C. Ten percent of the protein lysate was analyzed as the input. The remaining lysate (500 $\mu$g) was incubated with 1 $\mu$g V5 antibody or mouse IgG overnight at 4°C. Forty microliters of protein A/G-agarose was added to the mixture, and the mixture was then rotated for 2 h at room temperature. The beads were washed three times with 0.5% Tween 20 in cold PBS and resuspended in 40 $\mu$L 2× SDS loading buffer to elute conjugated proteins. Next, the beads were boiled at 95°C for 5 min and subjected to SDS-PAGE followed by immunoblot analysis as described above. In IP-mediated endogenous ubiquitination assays, cells were lysed with lysis buffer containing 5 $\mu$M ubiquitin aldehyde to inhibit deubiquitinating enzyme activity.

**Animal models.** The animal experiments under animal biosafety level 3 (ABSL3) were approved by the Institutional Animal Care and Use Committees at Tulane University. Transgenic *K18-hACE2* mice (*K18-hACE2$^{+/-}$*) (catalog number 034860) were obtained from the Jackson Laboratory that used the human cytokeratin 18 (*K18*) promoter to express human ACE2 in lung epithelial cells; nontransgenic *K18-hACE2* mice (*K18-hACE2$^{-/-}$*) were used as controls (50). SARS-CoV-2 isolate USA-WA1/2020, catalog number NR-52281, which was deposited at the Centers for Disease Control and Prevention and obtained through BEI Resources, NIAID, NIH, was used in the animal experiments. The virus was passaged in VeroE6 cells with Dulbecco's modified Eagle's medium (DMEM) complemented with 2% FBS, and the virus genome was sequenced for verification as previously described (20). The mice were intranasally infected with a moderate dose of SARS-CoV-2 (2 × 10$^5$ 50% tissue culture infective doses [TCID$_{50}$]) and euthanized at 3 and 6 days postinfection (dpi) under ABSL3.

**Immunofluorescence staining.** Lung paraffin sections from infected *K18-hACE2* transgenic (*K18-hACE2$^{+/-}$*) mice or *K18-hACE2$^{-/-}$* mice were subjected to immunofluorescence staining to detect the expression of proteins. The paraffin-embedded lung sections (5-$\mu$m thickness) were incubated overnight at 60°C. After deparaffinization in xylene, the slides were hydrated in a series of graded alcohols. Dewaxed slides were treated with Tris-EDTA buffer in a microwave oven for 20 min at 95°C and cooled to room temperature. The slides were incubated with an endogenous blocking solution (catalog number SP-6000-100; Vector Lab) for 10 min and with 2.5% normal horse serum for 20 min. The primary antibody (SARS-CoV-2 spike antibody [1:1,000] [catalog number 3425] from ProSci [Poway, CA]) was incubated overnight at 4°C, followed by incubation with goat anti-mouse IgG(H+L) Alexa Fluor 488-conjugated secondary antibody for 1 h. After counterstaining with 4′,6-diamidino-2-phenylindole (DAPI), images were digitally acquired by using the Zeiss Axio Scan Z1 system.

**COVID-19 patients.** Hospitalized COVID-19 patients from 4 April through 15 September 2020 were prospectively enrolled at the University of Pittsburgh Medical Center Presbyterian and Shadyside Hospitals. The study was approved by the Ethics Commission of the University of Pittsburgh and documented in the University of Pittsburgh Acute Lung Injury Registry and Biospecimen Repository (study protocol number 10050099) (51–53). Written informed consent was provided by all the participants or their legally authorized representatives. Exclusion criteria were preexisting chronic respiratory failure due to neuromuscular or neurological disease, the presence of tracheostomy, inability to obtain consent, prisoner status, comfort-measures-only status, and blood hemoglobin level of <8 g/dL.

**Enzyme-linked immunosorbent assay.** SARS-CoV-2-specific IgG responses in COVID-19 patients were analyzed by an indirect enzyme-linked immunosorbent assay (ELISA) as described previously (54). Flat-bottom microtiter plates were coated with 100 $\mu$L of diluted purified SARS-CoV-2 recombinant proteins (at a final concentration of 1,000 ng/mL) in 0.05 M carbonate-bicarbonate buffer (pH 9.6). Following three washes with 200 $\mu$L 0.1% (vol/vol) Tween 20 in PBS (PBS-T), the plate was blocked with 5% (wt/vol) nonfat milk in PBS for 1 h at 37°C. Human plasma samples were prepared at a 1:10 dilution in 1% (wt/vol) nonfat milk in PBS, and 100 $\mu$L of plasma was added to each well. The plate was washed three times and incubated with HRP-conjugated anti-human IgG at a dilution of 1:3,000 for 1 h at 37°C. After extensive washing, 100 $\mu$L of TMB substrate solution was added to each well and allowed to react for 10 min in the dark at 37°C. Finally, the reaction was stopped with 100 $\mu$L of 1 M $H_2SO_4$, and the absorbance of the optical density (OD) at 450 nm was read using an automated spectrophotometer. Moreover, we measured plasma IL-6 levels as part of a Luminex panel with a V-Plex human biomarker multiplex assay (MesoScale Diagnostics) as previously described (53).

**Statistical analyses.** All data are presented as means ± standard deviations (SD). All statistical analyses were performed using GraphPad Prism7 software. Continuous variables between two groups were compared by an unpaired *t* test. Pearson correlation coefficient tests were used to study the correlation between two variables. A *P* value of <0.05 was considered statistically significant.

## SUPPLEMENTAL MATERIAL

Supplemental material is available online only.

**FIG S1**, TIF file, 2.4 MB.

**FIG S2**, TIF file, 2.4 MB.

**TABLE S1**, DOCX file, 0.02 MB.

**TABLE S2**, DOCX file, 0.02 MB.

**TABLE S3**, DOCX file, 0.02 MB.

## ACKNOWLEDGMENTS

This work was supported by National Institutes of Health R01 grant HL142997 (C.Z.); career development award number IK2 BX004886 from the U.S. Department of Veterans Affairs Biomedical Laboratory R&D (BLRD) Service (W.B.); HL136143, HL142084, HL143285, and P01 HL114453 (J.S.L.); U01 HL098962 and K24 HL123342 (A.M.); K23 HL139987 (G.D.K.); and NIH 5 P51OD011104-58 and R21OD024931 (X.Q.).

We declare no conflict of interest with the manuscript.

C.Z. conceived the science and designed the experiments. W.L., T.L., K.V.F., and R.D. conducted immunoblot analysis. C.W. and X.Q. performed animal studies. G.D.K., W.B., A.M., and J.S.L. provided human plasma from COVID-19 patients and conducted IL-6 measurements. J.S.L. and X.Q. helped to develop the science and interpreted results. The manuscript was written by W.L. and C.Z. and edited by T.L., C.W., K.V.F., R.D., X.Q., G.D.K., W.B., J.S.L., and A.M. All authors read and approved the final manuscript.

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
