## [Reviewer comments · mSystems]

Stability of SARS-CoV-2-encoded proteins and their antibody levels correlation to IL-6 in COVID-19 patients

Wangyang Li, Georgios Kitsios, William Bain, Chenxiao Wang, Tiao Li, Kristen Fanning, Rushikesh Deshpande, Xuebin Qin, Alison Moris, Janet Lee, and Chunbin Zou

Corresponding Author(s): Chunbin Zou, University of Pittsburgh

Review Timeline:

Submission Date:	January 20, 2022
Editorial Decision:	March 15, 2022
Revision Received:	April 1, 2022
Accepted:	May 3, 2022

Editor: Aleksandra Nita-Lazar

Reviewer(s): Disclosure of reviewer identity is with reference to reviewer comments included in decision letter(s). The following individuals involved in review of your submission have agreed to reveal their identity: İkbâl Agah İnce (Reviewer #2)

Transaction Report:

DOI: <https://doi.org/10.1128/msystems.00058-22>

March 15, 2022

Dr. Chunbin Zou
University of Pittsburgh
Pittsburgh

Re: mSystems00058-22 (Stability of SARS-CoV-2-encoded proteins and their antibody levels correlation to IL-6 in COVID-19 patients)

Dear Dr. Zou:

Thank you for submitting your manuscript to mSystems. We have completed our review and I am pleased to inform you that, in principle, we expect to accept it for publication in mSystems. However, acceptance will not be final until you have adequately addressed the reviewer comments.

Preparing Revision Guidelines

Sincerely,

Aleksandra Nita-Lazar

Editor, mSystems

Journals Department
Reviewer comments:

Reviewer #1 (Comments for the Author):

In the manuscript entitled "Stability of SARS-CoV-2 encoded proteins and their antibody levels correlation to IL-6 in COVID-19 patients" Li et al. individually express predicted SARS-CoV-2 proteins and assess their half-life via SDS-PAGE. From this analysis they determined two subgroups of viral proteins: some which were more labile and some which were more stable. They went on to characterize how the proteins were degraded mechanistically implicating the proteasome. They subsequently assessed whether stability of the protein correlated with antibody levels to the protein from the serum of COVID-19 patients and tried to establish a correlation between levels of these antibodies and IL-6 levels.

Overall, I think that this a well written and interesting manuscript which will shed light on SARS-CoV-2 protein stability. I think it will also appeal to a wide readership. I have several minor suggestions to improve the manuscript.

Figure 6: It is unclear from the figure where each protein is being enriched and which cross reactive band is meant to be the ubiquitinated SARS-CoV-2 protein. This needs to be indicated with an arrow or a blot that shows the Coronavirus protein in parallel (i.e. V5 reprobe of the ubiquitin blot not just the input).

Figure 7: In the linear regression analysis, the long-lived proteins are plotted to have a half-life of 12 hours. Did the authors conduct experiments out to this timepoint or longer? In the earlier figures the last time point is at 8 hours. Is this linear regression sufficiently powered to draw the conclusion that the authors did?

Figure 8: Could the authors discuss best fit here and how that was decided and powered? It seems like four of the patients had extremely high IL-6 and variable levels other antibodies. If those patients are excluded would you draw the same conclusions? Are these four patients always the same and if so, does that really imply correlation? Could the authors discuss whether these data are sufficiently powered or whether this lends more information than the distribution analysis in Figure 7A?

In light of these limitations to the data (in particular Figure 7 and 8), I would suggest toning down the conclusions made on protein stability, antigen presentation and antibody titer (Lines 340-342) and on IL-6 (Lines 355-57) unless the authors wish to bolster these comments with more data. This reviewer felt that the way the data was handled in the discussion was more measured and the results should aim to match that tone.

Finally, did the authors consider using the mouse tissue/serum in Figure 1 to validate their findings in cell lines as to stability of proteins or in IL-6 versus antibody titers? The indirect ELISA could be useful here too and lend validation to their hypothesis since patient samples are (understandably) so limited.

Discussion points:

Is there anything known about the 19 patients in terms of severity of COVID or outcome of disease that could inform the analysis. The authors could discuss this.

Since some of the proteins from SARS-CoV-2 may stabilize each-other the authors could discuss this as a downside of expressing one protein at a time. The mouse data could of course complement this since it is from an infection where protein expression is under control of the virus. Another way to incorporate other groups data into their discussion and analysis would be to look at mass spec mapping of SARS-CoV-2 proteins in COVID patients (Nie X, et al. 2021 Multi-organ proteomic landscape of COVID-19 autopsies. Cell. 2021 Feb 4;184(3):775-791.e14. doi: 10.1016/j.cell.2021.01.004.).

Typographical errors:

Line 340: Linear instead of Liner

Reviewer #2 (Comments for the Author):

The study described a comprehensive study on individual stability of SARS-CoV-2-encoded proteins and their antibody levels correlation to IL-6 in COVID-19 patients. The presentation and used methodology are straightforward and very clear.

Several questions arising,

Why a complementation type of test was not considered as in viral infection is a dynamic interaction of viral proteins as a whole.

For example, the generation of an expression plasmid for ORF1ab that encode replicase polyprotein PP1ab and PP1a cleaving into 16 non-structural proteins (NSP) may provide a comparative picture for individual viral protein versus viral protein complementation impact on viral protein stability.

The second point that needs to be clarified is why only the C terminal fusion option is preferred?

Point by Point Response

Reviewer #1 (Comments for the Author):

In the manuscript entitled "Stability of SARS-CoV-2 encoded proteins and their antibody levels correlation to IL-6 in COVID-19 patients" Li et al. individually express predicted SARS-CoV-2 proteins and assess their half-life via SDS-PAGE. From this analysis they determined two subgroups of viral proteins: some which were more labile and some which were more stable. They went on to characterize how the proteins were degraded mechanistically implicating the proteasome. They subsequently assessed whether stability of the protein correlated with antibody levels to the protein from the serum of COVID-19 patients and tried to establish a correlation between levels of these antibodies and IL-6 levels.

Overall, I think that this a well written and interesting manuscript which will shed light on SARS-CoV-2 protein stability. I think it will also appeal to a wide readership. I have several minor suggestions to improve the manuscript.

Figure 6: It is unclear from the figure where each protein is being enriched and which cross reactive band is meant to be the ubiquitinated SARS-CoV-2 protein. This needs to be indicated with an arrow or a blot that shows the Coronavirus protein in parallel (i.e. V5 reprobe of the ubiquitin blot not just the input).

Answer: We thank the reviewer for the suggestive commentary. Each protein has been enriched by V5 immunoprecipitation for detection of ubiquitin (upper panel). The middle and lower panels are whole cell lysate, not enriched. Polyubiquitinated proteins are mostly degraded via proteasome in which the proteins are attached with a moiety of ubiquitin units. Polyubiquitinated proteins are usually seen as multiple bands (high molecular smear) in SDS-PAGE. According our experience, V5 re-probing may result in the similar results. We have labeled the poly-ubiquitinated proteins in the figure.

Figure 7: In the linear regression analysis, the long-lived proteins are plotted to have a half-life of 12 hours. Did the authors conduct experiments out to this timepoint or longer? In the earlier figures the last time point is at 8 hours. Is this linear regression sufficiently powered to draw the conclusion that the authors did?

Answer: We used protein synthesis inhibitor cycloheximide to study the half-life of the viral proteins. Cycloheximide itself decays in cell culture. In addition, cycloheximide treatment more than 8 h may lead to cell death. Thus, the reliability of the data observed by longer than 8 h experiments with additional cycloheximide is questionable. This is why we show the results in 8 h periods. In addition, we did the study for 24 h for several stable proteins including Spike protein and did not observe any degradation (data not shown). We believe that the half-life we obtained in this system is reliable. From the data presented in Figure 3, we could estimate the half-life of the stable proteins. As a matter of fact, the expected half-life of all the 8 stable proteins could be more than 24 h (Fig. 3C). However, we believe that the conclusion we draw is appropriate. To validate the conclusion, we need to conduct more experiments. Thus, we have toned down about the conclusion as the reviewer suggested.

Figure 8: Could the authors discuss best fit here and how that was decided and powered? It seems like four of the patients had extremely high IL-6 and variable levels other antibodies. If

those patients are excluded would you draw the same conclusions? Are these four patients always the same and if so, does that really imply correlation? Could the authors discuss whether these data are sufficiently powered or whether this lends more information than the distribution analysis in Figure 7A?

Answer: High level of IL6 has been reported in COVID-19 patients linked with poor prognosis. In our knowledge, the viral products that causes high IL-6 is not fully explored. IL-6 level in COVID patients could achieve 1000 pg/mL (Front. Immunol., 18 February 2021 <https://doi.org/10.3389/fimmu.2021.613422>). The highest level in our study is within 400 pg/mL. Thus, we believe that the data is in the range of actual laboratory examination, do not need to be excluded. According to our previous experience, when we use a system error $\alpha=0.05$, with a type 2 error of 20%, considering low standard deviation of the detection of IL-6 and the estimated effect size, the sample size we used should be satisfactory for this study. Due to the availability of the samples from patients, we were unable to conduct multiple time points of the patients, thereby we can only draw a conclusion from the limited data. The conclusion is preliminary but meaningful, needs further experimental evidence to validate in future studies, however, we think that it is beyond the scope of the current study. Therefore, we believe that the conclusion we draw is reasonable. In addition, we have toned down the conclusion following the reviewer's commentary.

In light of these limitations to the data (in particular Figure 7 and 8), I would suggest toning down the conclusions made on protein stability, antigen presentation and antibody titer (Lines 340-342) and on IL-6 (Lines 355-57) unless the authors wish to bolster these comments with more data. This reviewer felt that the way the data was handled in the discussion was more measured and the results should aim to match that tone.

Answer: We have followed your suggesting toned down the conclusions we drew.

Finally, did the authors consider using the mouse tissue/serum in Figure 1 to validate their findings in cell lines as to stability of proteins or in IL-6 versus antibody titers? The indirect ELISA could be useful here too and lend validation to their hypothesis since patient samples are (understandably) so limited.

Answer: We considered the suggested approaches from the reviewer when we started the experiments. Cell culture for SARS-CoV-2 infection needs BSL-3 level facility, which is not established in our department. We tried to seek extramural collaboration out of campus but failed. Further, the lack of validated antibodies against individual viral proteins is the biggest issue. The qualities and specificities of commercially available antibodies are not satisfactory even in ELISA as we showed (Fig. 7A). We tried to immunofluorescent stain mouse lung tissues with antibodies other than anti-Spike but was unsuccessful due to the non-specificity of the antibodies.

Discussion points:

Is there anything known about the 19 patients in terms of severity of COVID or outcome of disease that could inform the analysis. The authors could discuss this.

Answer: The general characters of the patients have been published in reference as we cited (Bain W, Yang H, Shah FA, Suber T, Drohan C, Al-Yousif N, DeSensi RS, Bensen N, Schaefer

C, Rosborough BR, Somasundaram A, Workman CJ, Lampenfeld C, Cillo AR, Cardello C, Shan F, Bruno TC, Vignali DAA, Ray P, Ray A, Zhang Y, Lee JS, Methe B, McVerry BJ, Morris A, Kitsios GD. 2021. COVID-19 versus Non-COVID-19 Acute Respiratory Distress Syndrome: Comparison of Demographics, Physiologic Parameters, Inflammatory Biomarkers, and Clinical Outcomes. *Ann Am Thorac Soc* 18:1202-1210). Our study is mainly focused on the protein stability of the virus. To better understand the severity and outcome of the disease with individual protein or antibody, we propose to investigate in future studies.

Since some of the proteins from SARS-CoV-2 may stabilize each-other the authors could discuss this as a downside of expressing one protein at a time. The mouse data could of course complement this since it is from an infection where protein expression is under control of the virus. Another way to incorporate other groups data into their discussion and analysis would be to look at mass spec mapping of SARS-CoV-2 proteins in COVID patients (Nie X, et al. 2021 Multi-organ proteomic landscape of COVID-19 autopsies. *Cell*. 2021 Feb 4;184(3):775-791.e14. doi: 10.1016/j.cell.2021.01.004.).

Answer: We have reworded the discussion by incorporating data from other groups. We have cited above recommended reference accordingly (Line 419).

Typographical errors:

Line 340: Linear instead of Liner

Answer: We have corrected it.

Reviewer #2 (Comments for the Author):

The study described a comprehensive study on individual stability of SARS-CoV-2-encoded proteins and their antibody levels correlation to IL-6 in COVID-19 patients. The presentation and used methodology are straightforward and very clear.

Several questions arising,

Why a complemantaiton type of test was not considered as in viral infection is a dynamic interaction of viral proteins as a whole.

For example, the generation of an expression plasmid for ORF1ab that encode replicase polyprotein PP1ab and PP1a cleaving into 16 non-structural proteins (NSP) may provide a comparative picture for individual viral protein versus viral protein complementation impact on viral protein stability.

Answer: *We thank the reviewer for the meaningful commentary. At the right beginning as we initiated the study, we were facing challenges to detect viral products. That is, how to use available tools to detect the degraded viral proteins in the cells. The availability of the validated antibodies against each of the viral proteins is very much limited. Most of the antibodies in the market are in poor quality with low specificity (please see the specificity in ELISA, Fig. 7A). As Reviewer 1 asked related questions, we tried to immunofluorescent stain other proteins in lung tissues but was not successful due to the poor quality of commercially available antibodies.*

Expression of the viral proteins with tag may give us a whole scenario in terms of protein stability as we presented in this study.

The second point that needs to be clarified is why only the C terminal fusion option is preferred?

Answer: *In our study, we used pcDNA™3.1 Directional TOPO® Expression Kit in which the tag is located at the C-terminus of insert. At the N-terminus, several elements have been included for directional and efficient expression of the insert. We have used this expression system for more than 10 years so when we started the study, we purchased this product without deep-thinking, we could use it more easily because of accumulated experience. Another reason is that we are studying protein stability, many proteins are degraded via N-terminus rule----- the composition of the amino acid residues at the N-terminus of a protein may affect its stability. Therefore, adding an N-terminal tag to a protein may artificially change the protein stability more likely.*

May 3, 2022

Dr. Chunbin Zou
University of Pittsburgh
Pittsburgh

Re: mSystems00058-22R1 (Stability of SARS-CoV-2-encoded proteins and their antibody levels correlation to IL-6 in COVID-19 patients)

Dear Dr. Chunbin Zou:

Your manuscript has been accepted, and I am forwarding it to the ASM Journals Department for publication. For your reference, ASM Journals' address is given below. Before it can be scheduled for publication, your manuscript will be checked by the mSystems production staff to make sure that all elements meet the technical requirements for publication. They will contact you if anything needs to be revised before copyediting and production can begin. Otherwise, you will be notified when your proofs are ready to be viewed.

Publication Fees:

We recognize that the video files can become quite large, and so to avoid quality loss ASM suggests sending the video file via <https://www.wetransfer.com/>. When you have a final version of the video and the still ready to share, please send it to mSystems staff at mSystems@asmusa.org.

For mSystems research articles, if you would like to submit an image for consideration as the Featured Image for an issue, please contact mSystems staff at mSystems@asmusa.org.

Sincerely,

Aleksandra Nita-Lazar

Editor, mSystems

Journals Department
Supplemental Figure 1: Accept
Table S1: Accept
Supplemental Figure 2: Accept
Table S2: Accept
Table S3: Accept